# COVID-19 and Intracranial Hemorrhage: A Multicenter Case Series, Systematic Review and Pooled Analysis

**DOI:** 10.3390/jcm11030605

**Published:** 2022-01-25

**Authors:** Moritz L. Schmidbauer, Caroline Ferse, Farid Salih, Carsten Klingner, Rita Musleh, Stefan Kunst, Matthias Wittstock, Bernhard Neumann, Karl-Michael Schebesch, Julian Bösel, Jana Godau, Piergiorgio Lochner, Elisabeth H. Adam, Kolja Jahnke, Benjamin Knier, Ingo Schirotzek, Wolfgang Müllges, Quirin Notz, Markus Dengl, Andreas Güldner, Oezguer A. Onur, Jorge Garcia Borrega, Konstantinos Dimitriadis, Albrecht Günther

**Affiliations:** 1Department of Neurology, University Hospital LMU Munich, 81377 Munich, Germany; moritz.schmidbauer@med.uni-muenchen.de (M.L.S.); stefan.kunst@med.uni-meunchen.de (S.K.); 2Department of Nephrology and Medical Intensive Care, Charité-Universitätsmedizin Berlin, 13353 Berlin, Germany; caroline.ferse@charite.de; 3Department of Neurology, Charité-Universitätsmedizin Berlin, 10117 Berlin, Germany; farid.salih@charite.de; 4Hans-Berger-Department of Neurology, Jena University Hospital, 07747 Jena, Germany; carsten.klingner@med.uni-jena.de (C.K.); rita.musleh@med.uni-jena.de (R.M.); 5Department of Neurology, Rostock University Hospital, 18147 Rostock, Germany; Matthias.wittstock@med.uni-rostock.de; 6Department of Neurology, Regensburg University, 93040 Regensburg, Germany; bernhard.neumann@medbo.de; 7Department of Neurology, Donau-Isar-Klinikum Deggendorf, 94469 Deggendorf, Germany; 8Medical Center, Department of Neurosurgery, University of Regensburg, 93053 Regensburg, Germany; karl-michael.schebesch@ukr.de; 9Department of Neurology, Klinikum Kassel, 34125 Kassel, Germany; julian.boesel@gnh.net (J.B.); jana.godau@klinikum-kassel.de (J.G.); 10Department of Neurology, Saarland University Medical Center, 66421 Homburg, Germany; piergiorgio.lochner@uks.eu; 11Department of Anesthesiology, University Hospital Frankfurt, 60590 Frankfurt, Germany; Elisabeth.Adam@kgu.de; 12Department of Neurology, University Hospital Frankfurt, 60590 Frankfurt, Germany; kolja.jahnke@kgu.de; 13Department of Neurology, Klinikum Rechts der Isar, Technical University of Munich, 81675 Munich, Germany; benjamin.knier@tum.de; 14Department of Neurology, Klinikum Darmstadt, 64283 Darmstadt, Germany; ingo.schirotzek@mail.klinikum-darmstadt.de; 15Department of Neurology, University Hospital Würzburg, 97070 Würzburg, Germany; Notz_q@ukw.de; 16Department of Neurosurgery, Carl Gustav Carus Medical University of Dresden, 01307 Dresden, Germany; markus.dengl@uniklinikum-dresden.de; 17Department of Anesthesiology, Carl Gustav Carus Medical University of Dresden, 01307 Dresden, Germany; andreas.gueldner@uniklinikum-dresden.de; 18Department of Neurology, University of Cologne, 50937 Cologne, Germany; oezguer.onur@uk-koeln.de; 19Department of Internal Medicine, University of Cologne, 50937 Cologne, Germany; jorge.garcia-borrega@uk-koeln.de; 20Institute for Stroke and Dementia Research (ISD), Ludwig-Maximilians-Universität (LMU), 81377 Munich, Germany

**Keywords:** COVID-19, intracranial hemorrhage, prognosis, anticoagulation

## Abstract

Introduction: Severe acute respiratory syndrome coronavirus 2 (SARS-CoV2) profoundly impacts hemostasis and microvasculature. In the light of the dilemma between thromboembolic and hemorrhagic complications, in the present paper, we systematically investigate the prevalence, mortality, radiological subtypes, and clinical characteristics of intracranial hemorrhage (ICH) in coronavirus disease (COVID-19) patients. Methods: Following the Preferred Reporting Items for Systematic Reviews and Meta-Analyses (PRISMA) guidelines, we performed a systematic review of the literature by screening the PubMed database and included patients diagnosed with COVID-19 and concomitant ICH. We performed a pooled analysis, including a prospectively collected cohort of critically ill COVID-19 patients with ICH, as part of the PANDEMIC registry (Pooled Analysis of Neurologic Disorders Manifesting in Intensive Care of COVID-19). Results: Our literature review revealed a total of 217 citations. After the selection process, 79 studies and a total of 477 patients were included. The median age was 58.8 years. A total of 23.3% of patients experienced the critical stage of COVID-19, 62.7% of patients were on anticoagulation and 27.5% of the patients received ECMO. The prevalence of ICH was at 0.85% and the mortality at 52.18%, respectively. Conclusion: ICH in COVID-19 patients is rare, but it has a very poor prognosis. Different subtypes of ICH seen in COVID-19, support the assumption of heterogeneous and multifaceted pathomechanisms contributing to ICH in COVID-19. Further clinical and pathophysiological investigations are warranted to resolve the conflict between thromboembolic and hemorrhagic complications in the future.

## 1. Introduction

Coronavirus disease 2019 (COVID-19), caused by severe acute respiratory syndrome coronavirus 2 (SARS-CoV2), manifests most commonly as a respiratory disease. However, a growing body of clinical data shows that neurological manifestations significantly contribute to the clinical spectrum of the disease and are especially relevant in critically ill patients [1,2,3].

In addition to other neurological manifestations, cerebrovascular disease has frequently been linked to acute SARS-CoV2 infection [4,5,6]. To account for the SARS-CoV2-associated hypercoagulable state, several pathophysiological mechanisms have been proposed, including both direct and indirect effects of the viral infection. Apart from hypercoagulable features, SARS-CoV2-associated endothelitis and microangiopathy are also postulated to contribute to hemorrhagic stroke [7,8,9]. Intracranial hemorrhage (ICH) in COVID-19 patients, therefore, might either be due to hemorrhagic transformation of ischemic stroke, primary hemorrhagic stroke, or traumatic ICH. Accordingly, the relevance of SARS-CoV2 related effects for the pathogenesis of these ICH subtypes might be heterogeneous.

As critically ill COVID-19 patients receive antithrombotic therapy in the majority of cases [10,11], including therapeutic range anticoagulation due to venous thromboembolism or extracorporeal membrane oxygenation (ECMO), the risk for ICH might be elevated. Additionally, the prevalence of ICH in these patients might be underestimated since prolonged sedation in COVID-19 patients potentially complicates valid neurological assessments to allow the timely detection of focal-neurological deficits.

So far, only scarce data are available on ICH in COVID-19 patients. The existing literature mainly consists of case reports and case series and thus does not allow further conclusions. Previous reviews in COVID-19 cohorts only covered intraparenchymal hemorrhages (IPH) [12], cerebrovascular disease in general [13], or reported quite low numbers of patients, leaving a significant gap in understanding the relevance of ICH in COVID-19.

To examine the prevalence, mortality, radiological subtypes, and clinical characteristics associated with ICH in COVID-19 patients, we combined a prospectively collected cohort of critically ill COVID-19 patients with individual and aggregate patient data from the literature and performed a pooled analysis.

## 2. Materials and Methods

### 2.1. Pandemic Registry

Cases with new ICH documented on computed tomography (CT) or magnetic resonance imaging (MRI) and simultaneous RT-PCR-confirmed infection with SARS-CoV2 were extracted from the prospective register study PANDEMIC (Pooled Analysis of Neurologic Disorders Manifesting in Intensive Care of COVID-19), which is conducted by the research network IGNITE (Initiative of German NeuroIntensive Trial Engagement) with support from the German Society for Neurologic Intensive Care and Emergency Medicine (DGNI). The PANDEMIC study aims to systematically elucidate neurologic manifestations in exclusively critically ill COVID-19 patients. Age; gender; stage of disease; the use of anticoagulation and, if stated, whether the therapeutic or prophylactic range was used, ECMO therapy, non-neurological symptoms and neurological symptoms, ICH subtype, time from COVID-19 diagnosis to ICH diagnosis, pertinent laboratory values, and modified Rankin Scale (mRS) at discharge and use of palliative care were extracted from a secured database and transferred to the data extraction sheet (S2). Particular disease phases of COVID-19 were defined according to the LEOSS registry (Lean European Open Survey for SARS-CoV-2 Infected Patients): 1. Uncomplicated phase without symptoms or slight symptoms of the upper respiratory tract, fever, or diarrhea; 2. Complicated phase when patients required oxygen supplementation and 3. Critical phase involving mechanical ventilation, dialysis and/or catecholamines [14]. Local ethics committees and institutional review boards of the participating centers approved the study based on the central vote of the ethics committee of Landesärztekammer Hessen, Germany (state medical association, 2020-1619-evBO, ethikkommission@laekh.de).

### 2.2. Systematic Review

#### 2.2.1. Eligibility Criteria

Recommendations given by the Preferred Reporting Items for Systematic Reviews and Meta-Analyses (PRISMA) statement were applied throughout the review. Methods of the analysis and inclusion criteria were specified in advance. The eligibility criteria were defined: types of participants (COVID-19 patients) and exposure (documented ICH during SARS-CoV2 infection). Non-English publications, repeat publications on the same cohort, publications on pediatric patients and publications with no full-text availability were excluded.

#### 2.2.2. Search Strategy

Studies were identified by searching the PubMed database as well as scanning the reference lists of articles. The last search was run on 5 March 2021. For the detailed search strategy, please refer to the Appendix A. The eligibility assessment was performed independently in an unblinded standardized manner by three authors (MLS, RM and KD). The disagreements between the reviewers were resolved by consensus. A data extraction sheet was designed, and the data was extracted accordingly. The information was extracted for the total number of patients with ICH and COVID-19; the overall number of patients at risk; age; sex; COVID-19 disease severity; COVID-19 associated symptoms; neurological symptoms; ICH subtype; anticoagulation; ECMO; time from COVID-19 diagnosis to ICH diagnosis; laboratory features; palliative care and mortality or mRS at discharge.

### 2.3. Statistical Analysis

First, we performed an analysis and provide descriptive statistics of the individual patient data (IPD) from both the PANDEMIC registry as well as single case reports in the literature. Regression with modified Rankin Scale (mRS) as the dependent variable was not performed due to incomplete data sets and the overall low patient numbers. To describe the characteristics of the complete cohort, we used the two-stage method [15] to combine IPD and aggregate patient data in published larger clinical studies. We classified data sets as IPDs whenever patient-specific data were available and used aggregate data when this was not the case. A pooled analysis using the inverse variance method was performed. If not stated otherwise, the values derived from the random effects model were used. The inconsistency of the data was measured using the method proposed by Higgins et al. [16] Here, I^2^ of 0% to 30% was considered as not important, 30% to 60% as moderate heterogeneity and 60% to 100% as substantial heterogeneity. We plotted the effect by the inverse of its standard error (‘funnel plot’) to assess the potential bias, both visually and formally with Egger’s test. Statistical analysis was performed using Microsoft Office Excel Version 365 (Version 16.57, Microsoft, Washington, DC, USA), SPSS (Version 28, IBM, New York, NY, USA) and R Studio (metafor package).

## 3. Results

### 3.1. Literature Screening and Article Selection

A literature search revealed a total of 217 citations. An additional three publications that met the criteria for inclusion were identified by checking reference lists. After excluding the duplicates, 214 hits remained. Of these, 107 did not meet the eligibility criteria after reviewing titles and abstracts, and 28 additional studies had to be removed after full-text review. Overall, a total of 79 studies were identified for inclusion in the review (Figure 1). These comprised of 26 studies with aggregate-level data (minimal patient number *n* = 3; maximum patient number *n* = 42) and 53 single case reports or case series with individual-level data. Together, data sets of 477 patients were available for analysis (34 individual patient-level data sets from the PANDEMIC registry, 108 individual-level data sets and 335 aggregate patient-level data sets from published reports).

### 3.2. Baseline Characteristics of Individual-Level Patient Data

Overall, a total of 34 patients from the PANDEMIC registry fulfilled the inclusion criteria. The demographic and clinical characteristics of patients from the PANDEMIC registry, as well as the results for all individual-level patient data (total *n* = 142, with *n* = 34 from the PANDEMIC registry and *n* = 108 individual-level data from the literature review), are shown in Table 1.

A vast majority of the patients showed unfavorable clinical outcomes at hospital discharge (mRS 3–6, n = 108/118, 91.5%). The clinical outcome (mRS 0–2 vs. 3–6) correlated with critical disease stage (20.0% vs. 66.7%, *p* = 0.001), time from COVID-19 diagnosis to ICH diagnosis (9.5 days vs. 16.0 days, *p* = 0.012), headache (40.0% vs. 11.5%, *p* = 0.012) and palliative care (0% vs. 38.6%, *p* = 0.015) (Table 2).

### 3.3. Pooled Analysis of Aggregate Data

The key results from the two-stage pooled analysis, including the individual-level patient data (n = 142 patients) as well as aggregate level data (n = 335 patients), are displayed in Table 3.

Here, the median age was 58.8 years (95% CI 54.8 years–62.9 years; I^2^ = 85.6%) and 34.0% patients were female (95% CI 29.5%–40.4%; I^2^ = 0%). A total of 23.3% patients experienced the critical stage of COVID-19 (95% CI 8.9%–61.2%, I^2^ = 53.8%). A total of 62.7% were on anticoagulation (95% CI 38.2%–103.0%, I^2^ = 82.6%) and 27.5% received ECMO (95% CI 5.8%–130.2%, I^2^ = 92.7%). The median time from COVID-19 diagnosis to diagnosis of ICH was 21.5 days (95% CI 14.9 days–28.0 days, I^2^ = 92.3%). The most frequently observed clinical symptoms were respiratory symptoms (60.9%, 95% CI 41.2%–90.0%, I^2^ = 64.0%) and an altered level of consciousness (57.3%, 95% CI 39.9%–82.3%, I^2^ = 45.0%). Microbleeds (51.1%, 95% CI 31.1%–84.2%, I^2^ = 85.1%), SAH (26.6%, 95% CI 16.8%–42.0%, I^2^ = 61.2%) and IPH (33.7%, 95% CI 23.3%–48.8%, I^2^ = 63.7%) were most frequently documented as ICH subtypes. The laboratory results show elevated CRP (228.1 mg/L; 95% CI 200.1 mg/L–256.0 mg/L; I^2^ = 0%), leukocytosis (13.1 × 10^9^/L, 95% CI 6.6 × 10^9^/L–19.5 × 10^9^/L, I^2^ = 97.4%) with a normal platelet count (222.9 × 10^9^/L, 95% CI 193.9 × 10^9^/L–251.8 × 10^9^/L; I^2^ = 63.9%) as well as moderately elevated INR (1.4, 95% CI 1.1–1.6; I^2^ = 93.7%). The aPTT was 45.5 s (95% CI 34.2 s–56.7 s; I^2^ = 98.3%). Data heterogeneity as described by I^2^ is moderate to high in most of the variables.

Twelve studies reported data on the prevalence of ICH in COVID-19 patients (Figure 2). Eleven studies reported mortality rates in patients with ICH and COVID-19 (Figure 3). The random effects model yielded a prevalence of 0.85% (95% CI 0.36%–1.99%; I^2^ = 98%) and a mortality of 52% (95% CI 40%–67%; I^2^ = 52%), respectively.

Data heterogeneity turned out high for the prevalence data (I^2^ = 98%) and moderate for the mortality data (I^2^ = 52%). To further explore this heterogeneity, a funnel plot was drawn. Regarding the data on mortality, the funnel plot appears symmetrical on visual inspection, and the Egger’s test result is non-significant (*p* = 0.80). However, the data on prevalence shows a pronounced horizontal scatter of effect estimates, with the Egger’s test being significant for asymmetry (*p* = 0.04) (Figure 4).

## 4. Discussion

This systematic review provides new insights into prevalence, mortality and key clinical features of ICH in COVID-19 patients.

### 4.1. Prevalence and Mortality

The calculated prevalence for ICH in COVID-19 patients in our study was low (0.85%, 95% CI 0.36%–1.99%; I^2^ = 98%), but moderately increased in comparison to a large cohort on hemorrhagic stroke in COVID-19 patients (0.3%; data from the French National Administrative database) [18]. More importantly, it was higher than the prevalence reported from the 2018–2019 seasonal influenza cohort (0.2%) [18]. However, a substantial amount of patients in our report suffered microbleeds, which are not mentioned and might have been missed in the French Administrative Database. Apart from the microbleeds, the frequent subtypes of ICH were IPH and SAH. Mortality in our study was 52% (95% CI 40%–67%; I^2^ = 52%), whereas the case-fatality rate for ICH in non-COVID-19 patients was reported at approximately 40% [19]. The case fatality rates for COVID-19 patients requiring invasive mechanical ventilation were found to be 45% in a large meta-analysis [20]. As only our cohort combines both potentially fatal diagnoses (ICH and COVID-19), higher mortality is reasonable.

### 4.2. Predictors of Clinical Outcome

On an individual patient level, the critical disease stage (20.0% vs. 66.7%, *p* = 0.001), time from COVID-19 diagnosis to ICH diagnosis (9.5 days vs. 16.0 days, *p* = 0.012), headache (40.0% vs. 11.5%, *p* = 0.014) and palliative care (0% vs. 38.4%, *p* = 0.016) correlated with outcomes at discharge (mRS 0–2 vs. mRS 3–6). The critical stage of COVID-19 and headaches in the context of IPH were previously described as predictors for worse outcomes [21,22]. However, in our study, headache predicted a better functional outcome (mRS 0–2). The discrepancy to already published data may be explained by a bias that could have developed because headache had been coded for both, COVID-19 and ICH. Although bleeding diathesis has been a fundamental factor in ICH in both COVID-19 and non-COVID-19 patients [23,24], and a significant proportion of patients had anticoagulation and showed changes in the respective biological biomarkers (aPTT, INR) in this cohort, we did not find anticoagulation to be a significant variable in our study. However, as we were not able to specify whether patients received prophylactic or therapeutic dose anticoagulation in a large fraction of the overall data, the effect on the outcome might be underestimated.

By pooled analysis of aggregate level data, we provided detailed descriptive statistics but refrained from meta-regression due to incomplete data sets and thus insufficient statistical power. The patients with ICH during active COVID-19 were predominantly male with a median age of 58.8 years (95% CI 54.8; 62.9). Basic epidemiological data are thus comparable to already published cohorts of COVID-19 [22,25]. The majority of patients experienced a critical phase of the disease, with respiratory symptoms and an altered level of consciousness being the dominant clinical features. The high proportion of patients with critical stage of COVID-19 is consistent with the studies reporting a relative increase in neurological symptoms with a more severe disease [2,26]. The median time from COVID-19 diagnosis to the diagnosis of ICH was 21.5 days (95% CI 14.9; 28.0), which might be due to the diagnostic difficulties in critically ill patients, or due to COVID-19-specific vasculopathy in the subacute stage of disease, or both. The high proportion of patients receiving ECMO further illustrates the severity of disease in this cohort. Yet, with a recent analysis reporting similar rates of IPH in COVID-19 and propensity score matched controls without COVID-19 [27], it appears unlikely that the viral infection is an independent risk factor further aggravating the already existing substantial risk of ICH during ECMO therapy. As ICH has been known to be a fatal complication of ECMO in COVID-19, as well as in other etiologies of Acute Respiratory Distress Syndrome (ARDS), cranial imaging should be encouraged in cases with neurological deterioration [28,29].

### 4.3. Pathophysiology

The pathomechanism behind the association of COVID-19 infection and ICH is still highly controversial, and many hypothetical constructs describing both direct and indirect effects of virus infection were discussed [8]. With neurotropism having been demonstrated, the potential direct mechanisms include the infection of vascular endothelium and consecutive endothelitis [7,30], as well as the downregulation of the angiotensin-converting enzyme 2 (ACE2), leading to elevated levels of angiotensin II with inflammation, increase in blood pressure and other deleterious downstream effects [31]. Hyperinflammatory syndrome with a loss of vascular integrity and disseminated coagulation are further described to play a role as indirect mechanisms [32,33].

Given that the different subtypes of ICH have a distinct pathophysiology, it appears plausible that the magnitude of the role of SARS-CoV2 described above varies. In this meta-analysis, we found a mixed pattern, with diffuse cerebral microbleeds affecting the juxtacortical deep white matter structures, including the corpus callosum, as well as the brainstem and cerebellum, to be the most prevalent subtype. Isolated deep microbleeds or lobar microbleeds typical for hypertensive angiopathy and cerebral amyloid angiopathy, respectively, were far less common. The diffuse pattern of microbleeds was previously recognized in COVID-19 as well as non-COVID-19 ARDS and is referred to as Critical Illness-Associated Cerebral Microbleeds (CIAMs) [34,35,36]. Additionally, mixed cerebral microbleeds are also frequently observed in patients with high-altitude cerebral edema [37]. When compared to the patients without microbleeds, respiratory failure is more pronounced [34,36,38,39]. For CIAM, data supports a multifaceted pathogenesis. Hypoxemia, uremia, microangiopathy, the formation of microthrombi and disseminated intravascular coagulation (DIC), or a combination of these factors, are discussed as significant drivers [34,36,37,38,39,40,41]. Two studies investigating microbleeds in COVID-19 suggested a potential role for COVID-19-associated microangiopathy, as patients with microbleeds showed thrombocytopenia and elevated D-Dimers [34,36]. Overall, it is impossible to state whether the observed bleedings are COVID-19-associated, rather than a phenomenon caused by the general pathophysiology and treatment strategies in critical illness.

Overall, the above-mentioned hypercoagulable features may predispose patients to thromboembolic complications in the venous and arterial circulation. However, with regard to ischemic stroke, epidemiological data do not show a higher prevalence among COVID-19 patients [18,42]. Thus, secondary ICH due to hemorrhagic transformation/parenchymal hematoma is likely to be dependent on the use of antithrombotic agents in stroke management rather than direct effects of SARS-CoV2. Sinus venous thrombosis is infrequently reported in the context of COVID-19, but may be driven by its hypercoagulable features [43]. Unfortunately, reports included too small patient numbers to deduct a reasonable prevalence.

Furthermore, only very few cases of SAH during COVID-19 have been described. In aneurysmal SAH, the hypothesis of arterial weakening by viral infection was eliminated decades ago. To date, there is no additional evidence that SARS-CoV2 could contribute to the pathogenesis of non-traumatic SAH [44]. As for IPH, a systematic review and pooled analysis revealed that 67.7% of patients exhibited atypical, lobar IPH, while in non-COVID-19 cohorts, proportions of 32% were reported [12,45]. COVID-19 patients with ICH had a multilocular manifestation of ICH in 20.6%, while others reported a prevalence of only about 6% for more heterogeneous cohorts [12,46]. In line with those findings of atypical localization, only 53% of patients had arterial hypertension. Overall, this suggests that additional factors may play a role in ICH in COVID-19 patients. As a majority receives anticoagulation, and a substantial proportion of critically ill-COVID-19 patients require ECMO therapy, it is likely that, in addition to the complex mechanisms already described above, those therapeutic interventions play a pivotal role for ICH in COVID-19. Indeed, therapeutic anticoagulation was found to increase the risk of IPH in COVID-19 by approximately 5-fold and was found to be a predictor of mortality [23].

### 4.4. Limitations

This review has several limitations. First, the data sets are incomplete due to a great heterogeneity of variables being reported, with detailed information for the cohort of interest often being unavailable in aggregate level data. Thus, a detailed analysis of ICH subtypes was only feasible using the individual data set. Second, and although inherent to any meta-analysis of special significance in this case, there are relevant sources of bias. As the pandemic is highly dynamic and often requires a rapid review and publication of scientific results, it is likely that data irregularities due to methodological issues or publication bias are particularly relevant in this research field. On the other hand, the limited resources during the pandemic with potentially limited access to health care facilities can lead to a substantial amount of undetected and underreported cases. With regard to the PANDEMIC cohort, neurological manifestations are frequent as neurological symptoms were mandatory for study inclusion. At the same time, the patients were often only screened later on in the course of the disease. Thus, the early non-specific symptoms, such as fever, might have been missed and are thus potentially underreported. Finally, the data on an adequate control group is not available.

### 4.5. Conclusions

In this systematic review and pooled analysis, we found intracranial hemorrhage to be associated with COVID-19 in 0.85% of the cases. ICH in COVID-19 patients is of poor prognosis, with 52% mortality. Given the potentially devastating consequences of ICH, cranial imaging in COVID-19 patients should be encouraged prior to ECMO to exclude ICH, and in any case with neurological deterioration. As ICH represents a heterogeneous entity, the magnitude of SARS-CoV2-associated effects most likely depends on the ICH subtype. More accurate epidemiological data and further pathophysiological insights are warranted to guide future clinical management.

## Figures and Tables

**Figure 1 jcm-11-00605-f001:**
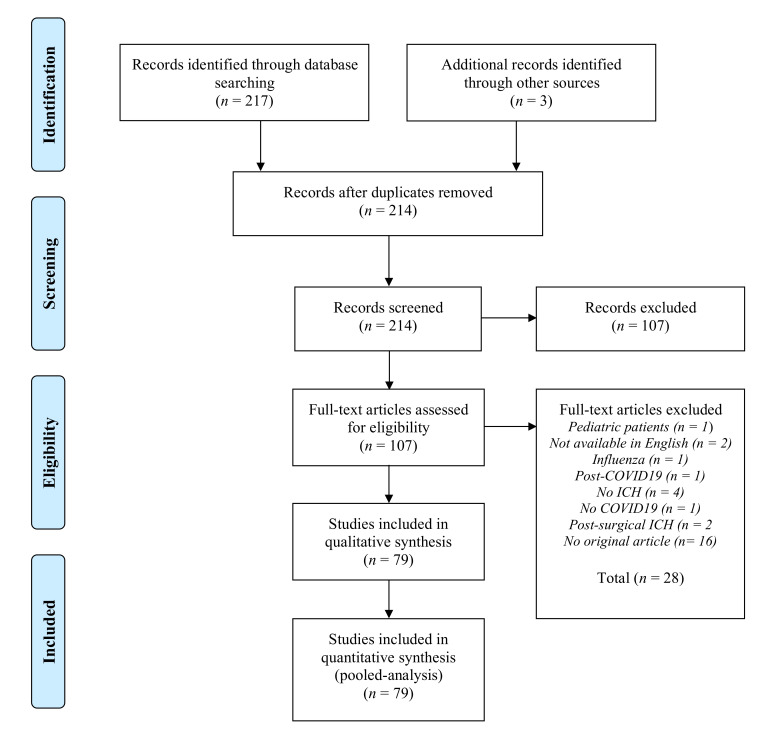
Study selection diagram adapted from the Preferred Reporting Items for Systematic Reviews and Meta-Analyses (PRISMA) group statement [17].

**Figure 2 jcm-11-00605-f002:**
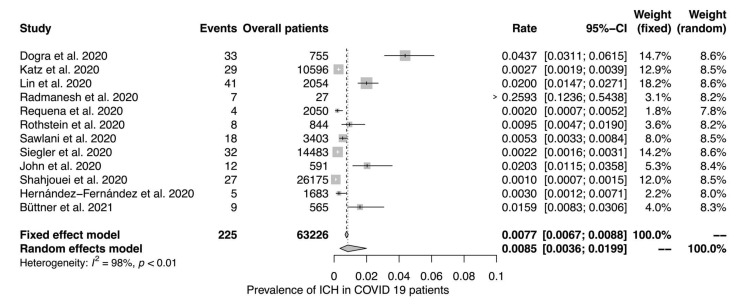
Prevalence of ICH in COVID-19 patients. The random and fixed effects model for the prevalence of ICH in COVID-19 patients. To account for the inconsistency in the measurements in the different studies, the effect estimates were calculated using the inverse variance method to allow for the weighing of the different variables according to their precision. The weight is illustrated by the size of the squares. Black and white lines indicate the respective 95% confidence intervals. The diamond shape indicates the average effect calculated by the fixed and random effects model, with the length of the shape representing the confidence interval.

**Figure 3 jcm-11-00605-f003:**
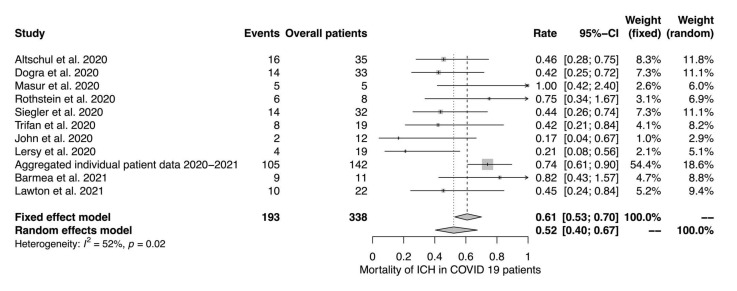
Mortality of ICH in COVID-19 patients. The random and fixed effects model for mortality of ICH in COVID-19 patients.

**Figure 4 jcm-11-00605-f004:**
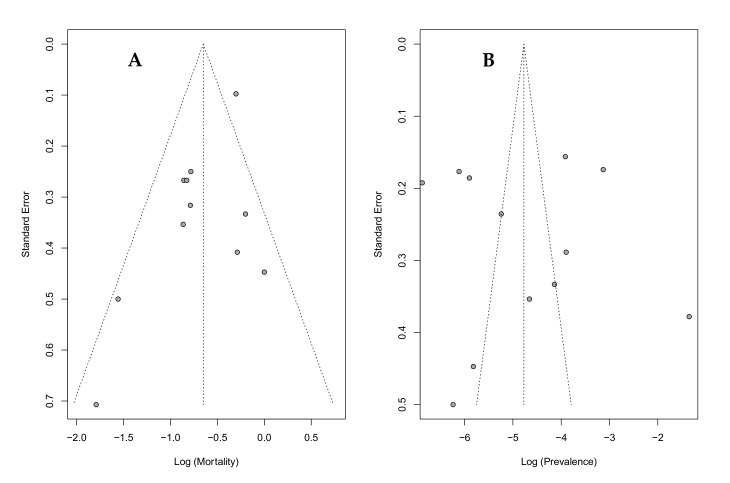
Heterogeneity and bias of studies reporting the prevalence and mortality of ICH in COVID-19 patients. Standard error as an indicator of study precision are plotted against study results ((**A**): mortality; (**B**): prevalence), with the vertical line representing the value derived from the random effects model. The diagonal lines indicate the corresponding 95% confidence intervals (‘funnel plot’). Each dot represents a single study. The Egger’s test for funnel plot asymmetry is significant for mortality (*p* = 0.04), but not for prevalence (*p* = 0.80).

**Table 1 jcm-11-00605-t001:** Baseline characteristics are displayed for the primary data from the PANDEMIC registry, and all individual data, respectively. The values are presented as the median with interquartile range (IQR) or as frequencies (*n*) and relative proportion of all the available valid data sets (%). Critical disease according to LEOSS (Lean European Open Survey on SARS-CoV-2 infected patients) registry criteria; ECMO extracorporeal membrane oxygenation; mRS modified Rankin Scale; COVID-19 coronavirus disease 2019; ICH intracranial hemorrhage; IPH intraparenchymal hemorrhage; SAH subarachnoid hemorrhage; IVH intraventricular hemorrhage; EDH/SDH epidural/subdural hematoma; HT/PH of IS hemorrhagic transformation/parenchymal hematoma of ischemic stroke; SVT sinus venous thrombosis; INR international normalized ratio and aPTT activated partial thromboplastin time.

Baseline Characteristics		PANDEMIC Registry (*n* = 34)	All Individual Patient Data (*n* = 142)
Age (years), median (IQR)		64.0 (57.0–76.0)	61.0 (53.8–71)
Female, *n* (%)		5/34 (14.7)	49/142 (34.5)
Critical disease (LEOSS), *n* (%)		22/34 (64.7)	81/128 (63.3)
ECMO, *n* (%)		12/34 (35.3)	21/101 (20.8)
Anticoagulation, *n* (%)		32/34 (94.1)	86/104 (82.7)
Time from COVID-19 diagnosis to ICH diagnosis (days), median (IQR)		21.0 (15.5–31.3)	15 (8.0–22.5)
Non-neurological symptoms	Fever, *n* (%)	3/34 (8.8)	50/119 (42.0)
	Respiratory symptoms, *n* (%)	20/34 (58.8)	89/116 (76.7)
	Myalgia/arthralgia, *n* (%)	3/34 (8.8)	10/119 (8.4)
	Malaise, *n* (%)	3/34 (8.8)	12/119 (10.1)
Neurological symptoms	Focal neurological deficits, n (%)	6/34 (17.6)	26/122 (21.3)
	Altered level of consciousness, *n* (%)	20/34 (58.8)	64/128 (50.0)
	Encephalopathy, *n* (%)	1/34 (2.9)	5/123 (4.1)
	Headache, *n* (%)	-	19/123 (15.5)
	Anisocoria, *n* (%)	12/34 (35.3)	29/124 (23.4)
	Seizure, *n* (%)	-	8/127 (6.3)
ICH	IPH, *n* (%)	20/34 (58.8)	68/142 (47.9)
	SAH, *n* (%)	16/34 (47.1)	44/142 (31.0)
	SDH/EDH, *n* (%)	4/34 (11.8)	8/142 (5.6)
	Microbleeds, *n* (%)	6/34 (17.6)	25/130 (19.2)
	Lobar microbleeds *n* (%)	-	1/25 (4.0)
	Deep microbleeds *n* (%)	2/6 (33.3)	2/25 (8.0)
	Mixed location microbleeds *n* (%)	4/6 (66.6)	22/25 (88.0)
	Not given *n* (%)	-	-
	IVH, *n* (%)	4/34 (11.8)	7/142 (4.9)
	HT/PH of IS, *n* (%)	-	3/142 (2.1)
	SVT with hemorrhage, *n* (%)	-	4/142 (2.8)
	Other, *n* (%)	-	-
	Multilocular ICH, not further specified, *n* (%)	-	-
Laboratory values	White blood cells (×10^9^/L), median (IQR)	20.3 (15.0–26.8)	15.8 (12.5–22.2)
	Platelet count (×10^9^/L), median (IQR)	121.5 (70.5–185)	176.0 (97.3–261.5)
	C-reactive protein (mg/L), median (IQR)	340.0 (231.0–402.0)	220.0 (54.5–340.0)
	INR, median (IQR)	1.4 (1.2–1.8)	1.3 (1.1–1.6)
	aPTT (s), median (IQR)	58.0 (44.0–75.0)	58 (38.8–68.0)
	D-dimer (mg/L), median (IQR)	17.9 (7.8–23.9)	6.8 (2.4–18.0)
mRS	0, *n* (%)	-	5/118 (4.2)
	1, *n* (%)	1/32 (3.1)	2/118 (1.7)
	2, *n* (%)	-	3/118 (2.5)
	3, *n* (%)	2/32 (6.3)	3/118 (2.5)
	4, *n* (%)	5/32 (15.6)	8/118 (6.8)
	5, *n* (%)	3/32 (9.4)	9/118 (7.6)
	6, *n* (%)	21/32 (65.6)	88/118 (74.6)
Death under palliative care, *n* (%)		13/24 (54.2)	34/101 (33.7)
Mortality	Total, *n* (%)	21/33 (64)	88/119 (73.9)
	IPH, *n* (%)	13/20 (65)	44/68 (64.7)
	SAH, *n* (%)	12/16 (75.0)	29/44 (65.9)
	SDH/EDH, *n* (%)	4/4 (100.0)	7/8 (87.5)
	Microbleeds, *n* (%)	3/6 (50.0)	13/25 (52.0)
	Lobar microbleeds *n* (%)	-	-
	Deep microbleeds *n* (%)	2/3	2/13
	Mixed location microbleeds *n* (%)	1/3	11/13
	Not given *n* (%)	-	-
	IVH, *n* (%)	2/4 (50.0)	3/7 (42.9)
	HT/PH of IS, *n* (%)	-	3/3 (100%)
	SVT with hemorrhage, *n* (%)	-	4/4 (100%)
	Other, *n* (%)	-	-
	Multilocular ICH, not further specified, *n* (%)	-	-

**Table 2 jcm-11-00605-t002:** The study cohort was dichotomized according to the outcome, with mRS 0–2 representing favorable and mRS 3–6 representing non-favorable outcomes. For a total of *n* = 118 patients, individual outcome data were available. Independent variables are presented as the median with interquartile range (IQR) or as frequencies (*n*) and relative proportion of all the available valid data sets (%). The Mann–Whitney U, chi-square and Fisher’s exact test were used as statistical tests. The results were considered statistically significant if the *p*-value was below 0.05.

Baseline Characteristics		Favorable Outcome (mRS 0–2)	Non-Favorable Outcome (mRS 3–6)	*p*-Value
Age (years), median (IQR)		60.5 (43.25–67.25)	60.0 (53.0–71.0)	0.329
Female, *n* (%)		5/10 (50.0)	38/108 (35.2)	0.630
Critical disease (LEOSS), *n* (%)		3/10 (20.0)	72/108 (66.7)	0.001
ECMO, *n* (%)		-	17/80 (21.3)	0.147
Anticoagulation, *n* (%)		3/5 (60.0)	77/88 (87.5)	0.085
Time from COVID-19 diagnosis to ICH diagnosis (days), median (IQR)		9.5 (1.8–13.5)	16.0 (10.0–24.5)	0.012
Non-neurological symptoms	Fever, *n* (%)	5/9 (55.6)	35/94 (31.8)	0.281
	Respiratory symptoms, *n* (%)	7/9 (77.8)	69/91 (75.8)	0.896
	Myalgia/arthralgia, *n* (%)	0/9 (10.0)	8/94 (8.5)	0.362
	Malaise, *n* (%)	2/9 (22.2)	8/94 (8.5)	0.184
Neurological symptoms	Focal neurological deficits, *n* (%)	2/10 (20.0)	22/96 (22.9)	0.834
	Altered level of consciousness, *n* (%)	3/10 (10.0)	48/101 (47.5)	0.289
	Encephalopathy, *n* (%)	-	4/97 (3.7)	0.513
	Headache, *n* (%)	4/10 (40.0)	11/96 (11.5)	0.014
	Anisocoria, *n* (%)	-	28/98 (28.6)	0.050
	Seizure, *n* (%)	1/10 (10.0)	7/99 (7.1)	0.735
ICH	IPH, *n* (%)	2/10 (20.0)	56/108 (51.9)	0.054
	SAH, *n* (%)	4/10 (40.0)	33/108 (30.6)	0.538
	SDH/EDH, *n* (%)	-	7/108 (6.4)	0.406
	Microbleeds, *n* (%)	3/10 (30.0)	18/102 (17.6)	0.340
	Lobar microbleeds *n* (%)	1/3 (33.3)	-	0.089
	Deep microbleeds *n* (%)	-	2/18 (11.1)	1.0
	Mixed location microbleeds *n* (%)	2/3 (66.6)	16/18 (88.9)	0.662
	Not given *n* (%)	-	-	-
	IVH, *n* (%)	1/10 (10.0)	4/108 (3.7)	0.334
	HT/PH of IS, *n* (%)	-	2/108 (1.9)	0.664
	SVT with hemorrhage, *n* (%)	-	4/108 (3.7)	0.536
	Other, *n* (%)	-	-	-
	Multilocular ICH, not further specified, *n* (%)	-	-	-
Laboratory values	White blood cells (×10^9^/L), median (IQR)	17.8 (NA)	16.0 (10.9–22.5)	0.758
	Platelet count (×10^9^/L), median (IQR)	103.0 (NA)	167.5 (94.3–244.5))	0.537
	C-reactive protein (mg/L), median (IQR)	142.0 (49.8–332.6)	230.0 (55.0–352.5)	0.457
	INR, median (IQR)	1.2 (NA)	1.3 (1.1–1.6)	0.457
	aPTT (s), median (IQR)	115.0 (NA)	56.0 (37.6–68.0)	0.117
	D-dimer (mg/L), median (IQR)	7.6 (NA)	7.8 (2.4–19.7)	0.550
Palliative care, *n* (%)		-	34/86 (38.4)	0.016

**Table 3 jcm-11-00605-t003:** Pooled analysis of the baseline characteristics, including all the available aggregate data. Individual data was aggregated prior to this analysis according to the two-stage method and was subsequently counted as one study. The number of total studies included in the analysis is displayed. The means and the 95% confidence interval are derived from the random effects model. The level of heterogeneity was expressed by I^2^ together with the 95% confidence interval.

Baseline Characteristics		Mean (95%-CI)	I^2^ (95%-CI)	Number of Studies
Age (years)		58.8 (54.8; 62.9)	85.6% (75.9%; 91.4%)	11
Female (%)		34.0 (29.5; 40.4)	0.0% (0.0%; 36.0%)	14
Critical disease (LEOSS) (%)		23.3 (8.9; 61.2)	53.8% (0.0%; 83.0%)	5
ECMO (%)		27.5 (5.8; 130.2)	92.7% (82.0%; 97.0%)	3
Anticoagulation (%)		62.7 (38.2; 103.0)	82.6% (55.3%; 93.2%)	4
Time from COVID-19 diagnosis to ICH diagnosis (days)		21.5 (14.9; 28.0)	92.3% (86.0%; 95.8%)	6
Non-neurological symptoms	Fever (%)	36.6 (19.9; 63.5)	71.0% (17.3%; 89.9%)	4
	Respiratory symptoms (%)	60.9 (41.2; 90.0)	64.0% (0.0%; 87.8%)	4
	Myalgia/arthralgia (%)	7.0 (3.8; 13.1)	NA	1
	Malaise (%)	8.5 (4.8; 14.9)	NA	1
Neurological symptoms	Focal neurological deficits (%)	23.8 (16.8; 33.8)	16.4% (0.0%; 82.6%)	5
	Altered level of consciousness (%)	57.3 (39.9; 82.3)	45.0% (0.0%; 79.8%)	5
	Encephalopathy (%)	24.4 (7.4; 80.1)	90.5% (78.7%; 95.8%)	4
	Headache (%)	13.9 (8.5; 20.1)	0% (NA)	2
	Anisocoria (%)	20.4 (14.2; 29.4)	NA	1
	Seizure (%)	8.4 (4.6; 15.4)	21.6% (0.0%; 67.1%);	5
ICH	IPH (%)	33.7 (23.2; 48.8)	63.7% (30.6%; 81.0%)	11
	SAH (%)	26.6 (16.8; 42.0)	61.2% (27.2%; 79.3%)	12
	SDH/EDH (%)	12.6 (4.4; 35.9)	84.0% (66.7%; 92.3%)	6
	Microbleeds (%)	54.7 (34.4; 87.1)	84.5% (73.8%; 87.1%)	11
	Lobar microbleeds	1.3 (0.3–5.2)	NA	2
	Deep microbleeds	8.8 (2.3–28.2)	72.2 (30.1–89.0)	5
	Mixed location microbleeds	35.6 (19.3–65.6)	82.4 (66.5–90.7)	8
	Not given	44.7 (29.5–67.7)	47.4 (0.0–79.2)	6
	IVH (%)	5.9 (3.0; 11.6)	2.7% (NA)	2
	HT/PH of IS (%)	9.2 (2.2; 39.1)	80.0% (36.7%; 93.7%)	3
	SVT with hemorrhage (%)	2.9 (1.2; 7.1)	0% (NA)	2
	Other (%)	46.2 (25.2; 84.8)	76.4% (50.3%; 88.7%)	7
	Multilocular ICH, not further specified (%)	23.0 (6.1; 0.87.0)	83.8% (59.0%; 93.6%)	4
Laboratory values	White blood cells (×10^9^/L)	13.1 (6.6; 19.5)	97.4% (95.5%; 98.5%)	4
	Platelet count (×10^9^/L)	222.9 (193.9; 251.8)	63.9% (12.7%; 85.1%)	6
	C-reactive protein (mg/L)	228.1 (200.1; 256.0)	0% (NA)	2
	INR	1.4 (1.1; 1.6)	93.7% (87.0%; 96.9%)	4
	apTT (s)	45.5 (34.2; 56.7)	98.3% (97.2%; 98.9%)	4
	D-dimer (mg/L)	8.2 (1.8; 14.6)	98.1% (96.5%; 99.0%)	3

## Data Availability

All data generated or analyzed during this study are included in this published article (and its Appendix A).

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
