# Peer review of "COVID-19 and Intracranial Hemorrhage: A Multicenter Case Series, Systematic Review and Pooled Analysis"

_jcm, 2022, doi:10.3390/jcm11030605_

Round 1

Reviewer 1 Report

The manuscript contains results of a multicenter cooperative study on the incidence and consequences of intracerebral hemorrhage among Covid-19 patients, depending on the advancement of the disease. Prospective data of individual patients from an all-German database were matched to personalized data from case-studies published in the literature and then combined with pooled data from published case series. The literature was searched under rigorous PRISMA regimen which makes the material rather credible.

For elaboration of the obtained data an advanced, laborious statistics were used which is for an average reader a bit difficult to follow  but as a fact supports the conclusions.

The prognosis of ICH in COVID-19 infection revealed  rather poor but luckily, the incidence of this serious condition is not so high (though somewhat higher than in some previous studies, e.g. French).

I have got some minor remarks.

Tables and figures must be of course self-explaining but definitions of all acronims with each subsequent table seems an exaggeration of this rule.

Figures are in turn somewhat unclear and difficult to follow. Some explanations might be of interest for a reader who is not deeply involved in statistics. For example: What a value is represented by time? What is the meaning of black and white arrow? Generally, the authors could put some effort in  trying to formulate their statistical conclusions in a bit more accessible, i.e. descriptive  language instead of displaying "in crudo" the work of the statistician.

Conclusions: lines 389-390 twice repeated "poor prognosis"

Reviewer 2 Report

The authors performed a pooled analysis with 79 literatures which included 477 patients with COVID-19 and intracranial hemorrhage (ICH) by screening the PubMed database, and found that incidence of ICH in the patients with COVID-19 was 0.85% and mortality of that 52.18%. ICH in COVID-19 patients is rare, but it has a very poor prognosis.

There are several critical queries for the acceptance. The authors have to address each problem one by one.

Table 1:

#1. Most of the figures for % in the brackets are in [%]. For example, 5/34 (14.7) should be corrected to 5/34 [14.7] in Female of PANDEMIC registry.

#2. Definition of critical disease is not clear. Please indicate it.

#3. In PANDEMIC registry, the number of patients with fever was too low (3, 8.8%). On the other hand, the number of patients with focal neurological deficits, altered level of consciousness, and ICH was higher than that of the patients with fever. Ordinally, I believe that the patients with focal neurological deficits, altered level of consciousness, and ICH associated with COVID-19 should have suffered from fever in acute phase, and subsequently experience these symptoms above and ICH. I guess there is a certain contradiction. Please check the numbers were correct, or explain the enigma in Discussion section.

#4. In the line of mRS, PANDEMIC registry has only 32 patients (=1+2+5+3+21), however the denominator was 33.

Table 2.

#5. Table 1 indicated that the numbers of patients with fever, IPH, SAH, SDH/EDH, and IVH were counted among all 176 patients in PANDEMIC registry (n=34), and all individual patient data (n=142). However, Table 2 showed 5/9+35/96 in fever, 2/10+57/110 in IPH…etc, to put it simply the denominator of each item above must have been 176.

Table 3.

#6. The incidence of microbleeds was 51.1%. Basically, microbleeds was divided into three categories consisting of deep CMBs, lobar CMBs, and mixed CMBs according to locations, and each category is believed that it has different causes such as hypertension, or cerebral amyloid angiopathy… Please classify CMBs into deep CMBs, lobar CMBs, and mixed CMBs, furthermore consider the pathogenesis based on the distribution of CMBs in Discussion.

Round 2

Reviewer 2 Report

The manuscript has been amended and is well written.